# Race and Outcomes to [^177^Lu]Lu-PSMA-617 in Advanced Prostate Cancer

**DOI:** 10.3390/cancers17121960

**Published:** 2025-06-12

**Authors:** Avina Rami, Caiwei Zhong, Miguel Muniz, Wanling Xie, Adam Khorasanchi, John Gallagher, Sedra Mohammadi, Daniel Fein, Andrew F. Voter, Hailey Stoltenberg, Dharmesh Gopalakrishan, Yuanquan Yang, Thomas S. C. Ng, Andrei Gafita, Daniel S. Childs, Heather Jacene, Praful Ravi

**Affiliations:** 1Dana-Farber Cancer Institute, Boston, MA 02215, USA; 2Mayo Clinic, Rochester, MN 55905, USA; 3Division of Hospital Medicine, The Ohio State University Comprehensive Cancer Center, Columbus, OH 43210, USA; 4Roswell Park Comprehensive Cancer Center, Buffalo, NY 14203, USA; 5Massachusetts General Hospital, Boston, MA 02214, USA; 6Beth Israel Deaconess Medical Center, Boston, MA 02215, USA; 7Johns Hopkins University School of Medicine, Baltimore, MD 21205, USA; 8Brigham & Women’s Hospital, Boston, MA 02115, USA

**Keywords:** [^177^Lu]Lu-PSMA-617, beta-emitter, castration-resistant prostate cancer, race, disparities

## Abstract

Black men with prostate cancer are more likely to die from their disease than White men. One of the newest treatments for advanced prostate cancer is a radioactive therapy, LuPSMA, that targets cancer cells directly. However, little is known about how well this treatment works for patients of different racial backgrounds. In this study, we looked at over 650 patients treated with LuPSMA across multiple hospitals in the United States and compared their outcomes by race. We found that Black and White patients had similar responses to the treatment and lived for a similar amount of time after receiving it. These results suggest that LuPSMA is equally effective across racial groups. Our findings highlight the need to make sure that all patients—regardless of race—can access the latest treatments. Efforts should continue to improve equity in healthcare and to include more diverse patients in clinical research.

## 1. Introduction

Prostate cancer disproportionately affects Black men, who are more likely to be diagnosed with metastatic disease and have higher mortality rates compared to non-Hispanic white men [1]. This disparity underscores the importance of evaluating treatment efficacy across different racial groups. Black and minority patients have generally been underrepresented in clinical trials, leading to inadequate data on the efficacy and safety of therapies for metastatic castration-resistant prostate cancer (mCRPC) in these populations.

Prostate-specific membrane antigen (PSMA) is a transmembrane glutamate carboxypeptidase highly expressed on metastatic prostate cancer cells. [^177^Lu]Lu-PSMA-617 (LuPSMA) is a radionuclide targeting PSMA-positive cells, delivering beta-particle radiation to both cancer cells and their surrounding microenvironment. In the phase 3 VISION trial, LuPSMA led to a median 4-month improvement in overall survival (OS) compared to best supportive care in men with mCPRC who had previously received taxane chemotherapy and an androgen receptor pathway inhibitor (ARPI) [2]. More recently, LuPSMA demonstrated an improvement in radiographic progression-free survival (rPFS) compared to ARPI switch in the pre-chemotherapy mCRPC setting [3].

Prior research has suggested that Black patients with mCRPC may experience greater benefit from proven life-prolonging therapies such as abiraterone, enzalutamide, and radium-223 compared to non-Hispanic white patients [4,5,6]. Other studies have shown that Black and other minority patients derive equal benefit from taxane chemotherapy and androgen receptor pathway inhibitors (ARPIs) for mCRPC as White patients [7,8]. However, the impact of race on outcomes with LuPSMA therapy remains unexplored, with a total of only 51 non-White patients (including 41 Black patients) randomized to receive LuPSMA as part of the two largest phase 3 trials reported to date [2,3]. Based on these considerations, we aimed to evaluate the association between clinical outcomes with LuPSMA and race.

## 2. Patients and Methods

This retrospective cohort study utilized electronic health record-derived deidentified patient data from seven institutions across the United States; Institutional Review Board approval was obtained at each participating center. Included patients had mCRPC, had previously been treated with at least one taxane and one ARPI, and received at least 2 cycles of LuPSMA between December 2021 and February 2024, either as standard-of-care or as part of an expanded access program. Only patients who received ≥2 cycles of therapy were included to permit assessment of PSA response. A standardized datasheet was used to collect relevant demographic and clinical variables, including self-reported race, from each institution. Patients were categorized into racial groups as follows: Black, Hispanic/Latino, Asian, and non-Hispanic White; Hispanic and Asian patients were considered to be non-Black minorities (NBM).

The key outcome measures were PSA-50 (defined as the proportion of patients achieving a ≥50% decrease in PSA at any point during LuPSMA therapy), PSA progression-free survival (PSA-PFS, defined as the duration from the start of LuPSMA therapy to the date of PSA progression, death, or censored at the last PSA evaluation; PSA progression was defined as a 25% increase in PSA levels above nadir or from baseline if no PSA decline, along with an absolute increase of at least 2 ng/mL), and overall survival (OS, measured from the start of LuPSMA therapy to the date of death or censored at last follow-up).

Baseline patient information and treatment cycles were summarized and compared across all patients and by racial categories. Continuous variables were analyzed using t-tests, while categorical variables were assessed using chi-square tests. For categorical variables with expected cell counts less than five, Fisher’s exact test was applied. PSA-PFS and OS were estimated using the Kaplan–Meier method. Logistic regression was used to estimate odds ratios (ORs) and 95% confidence intervals (CIs) for PSA-50 by racial group. Cox proportional hazards models were applied to examine the association of race with PSA-PFS and OS, adjusting for age at treatment initiation, number of prior systemic therapies, sites of metastasis, and baseline PSA levels (log-transformed). All statistical analyses were performed using SAS version 9.4 (SAS Institute, Cary, NC, USA).

## 3. Results

Baseline characteristics of the 654 patients included are shown in Table 1. Overall, 593 (91%) self-identified as White, 45 (7%) as Black, and 16 (2%) as NBM (8 were Asian and 8 were Hispanic). Black patients had significantly higher baseline PSA values (median 137 vs. 32 ng/mL, *p* < 0.01) as well as a higher incidence of bone metastases (98% vs. 88%, *p* = 0.05) compared to White patients, though both groups had a similar burden of visceral metastases (20% vs. 23%, *p* = 0.65). Median follow-up in alive patients was 9.1 months (95% CI 6.9–12.0).

The likelihood of achieving a PSA-50 was similar between White (51% (47–55)), Black (49% (33–65), adjusted OR = 0.86 (0.45–1.30)) and NBM patients (56% (30–80), adjusted OR = 1.30 (0.47–3.71)). Table 2 summarizes PSA-PFS and OS data across racial groups, and Kaplan–Meier curves for PSA-PFS and OS by race are shown in Figure 1. While median PSA-PFS was numerically lower for Black patients (4.4 months (2.8–NR)) compared to White (7.1 (6.9–7.8)) or NBM patients (7.3 (1.5–8.8)), this was not significant on adjusted analyses (Black: adjusted HR = 1.14 (0.72–1.81); NBM: adjusted HR = 1.16 (0.57–2.35)). Similarly, while there was a trend towards inferior OS in Black patients (12-month OS 46% (29–66) vs. 61% (56–77) for White and 75% (47–95) for NBM), this was not significant on adjusted analyses (Black: adjusted HR = 1.37 (0.84–2.23); NBM: adjusted HR = 0.68 (0.27–1.74)). Similar results were seen when comparing White patients to all non-White patients (Appendix A).

## 4. Discussion

In this retrospective multicenter analysis of more than 650 patients with mCRPC treated with LuPSMA at seven institutions across the U.S., we noted that clinical outcomes to therapy were comparable between Black and White patients, specifically with regard to PSA response, PSA-PFS, and OS. To our knowledge, this is the first study evaluating the association between race and outcomes with LuPSMA in prostate cancer. While our cohort of Black men is small (n = 45), it is an informative addition to the literature since only 41 Black patients were randomized to LuPSMA on the VISION and PSMAfore trials [2].

Our results align with prior studies that have similarly found no significant racial differences—and perhaps even improved outcomes in Black patients—in patients with mCRPC treated with taxane chemotherapy or ARPIs. A pooled analysis of nearly 9000 patients treated on phase 3 trials evaluating docetaxel in mCRPC showed no differences in outcomes by race, while a trend towards improved outcomes in Black patients in mCRPC was seen in another study performed in a healthcare setting with equal access to care [9]. Similar results have been observed in post hoc analyses of a randomized trial evaluating an ARPI in the hormone-sensitive setting [8], while a prospective trial evaluating abiraterone in Black and White mCRPC patients showed similar efficacy between the two groups [10]. The totality of the data clearly shows that differences in therapeutic response do not underlie the overall higher prostate cancer mortality observed in Black patients in population-based studies [11].

From a public health perspective, our findings underscore the importance of ensuring equitable access to LuPSMA therapy for all patients with mCRPC, particularly since this is the newest therapy approved for prostate cancer. Evidence from other settings supports the notion that racial disparities in prostate cancer outcomes are largely attributable to systemic differences in access to care. For example, a study comparing the SEER registry and the equal-access Veterans Health Administration (VHA) found that racial disparities in prostate cancer-specific mortality were evident in SEER but not in the VHA, largely due to differences in the stage of disease at presentation. These findings suggest that improving access to timely diagnosis and treatment may help bridge disparities in survival [12]. Nevertheless, delays in care persist: non-White men with localized prostate cancer face significantly higher odds of major treatment delays, which have worsened in the years following implementation of the Affordable Care Act [13]. These data emphasize the need for healthcare systems to address structural and interpersonal barriers to equitable cancer care delivery. Policy efforts should focus on addressing barriers to access that disproportionately affect minority patients, such as insurance coverage, treatment availability, and potential healthcare provider biases [14]. Furthermore, the prostate cancer community as a whole must strive to ensure that clinical trials reflect the diversity of prostate cancer patients globally.

While our study provides valuable insights, it is not without limitations. The overall sample size for Black and NBM patients was relatively small and we relied on self-reported race. Our analysis may also not fully account for genomic or socio-economic factors or healthcare access issues that could influence treatment outcomes. Data on other established prognostic factors [15], such as hemoglobin and number of lesions at baseline, were not available, though our analyses accounted for sites of metastases and number of prior therapies. We did not evaluate rPFS as an outcome given the heterogeneity in practice with regard to radiographic monitoring of patients on therapy, challenges in investigator assessment of bone scan findings and lack of consensus criteria on PET response in mCRPC; instead, we focused on PSA-50 and OS given that these are clearly discernible endpoints available for all patients. Further studies, ideally from geographically diverse populations, are needed to validate our findings.

## 5. Conclusions

In conclusion, we noted similar efficacy of LuPSMA therapy for Black and White patients with mCRPC, contributing to the growing evidence that prostate cancer therapies offer equal treatment benefits across racial groups. These findings have implications for clinical practice and public health policy, and further work to broaden access to care and evaluate prostate cancer therapies in diverse patient populations is needed.

## Figures and Tables

**Figure 1 cancers-17-01960-f001:**
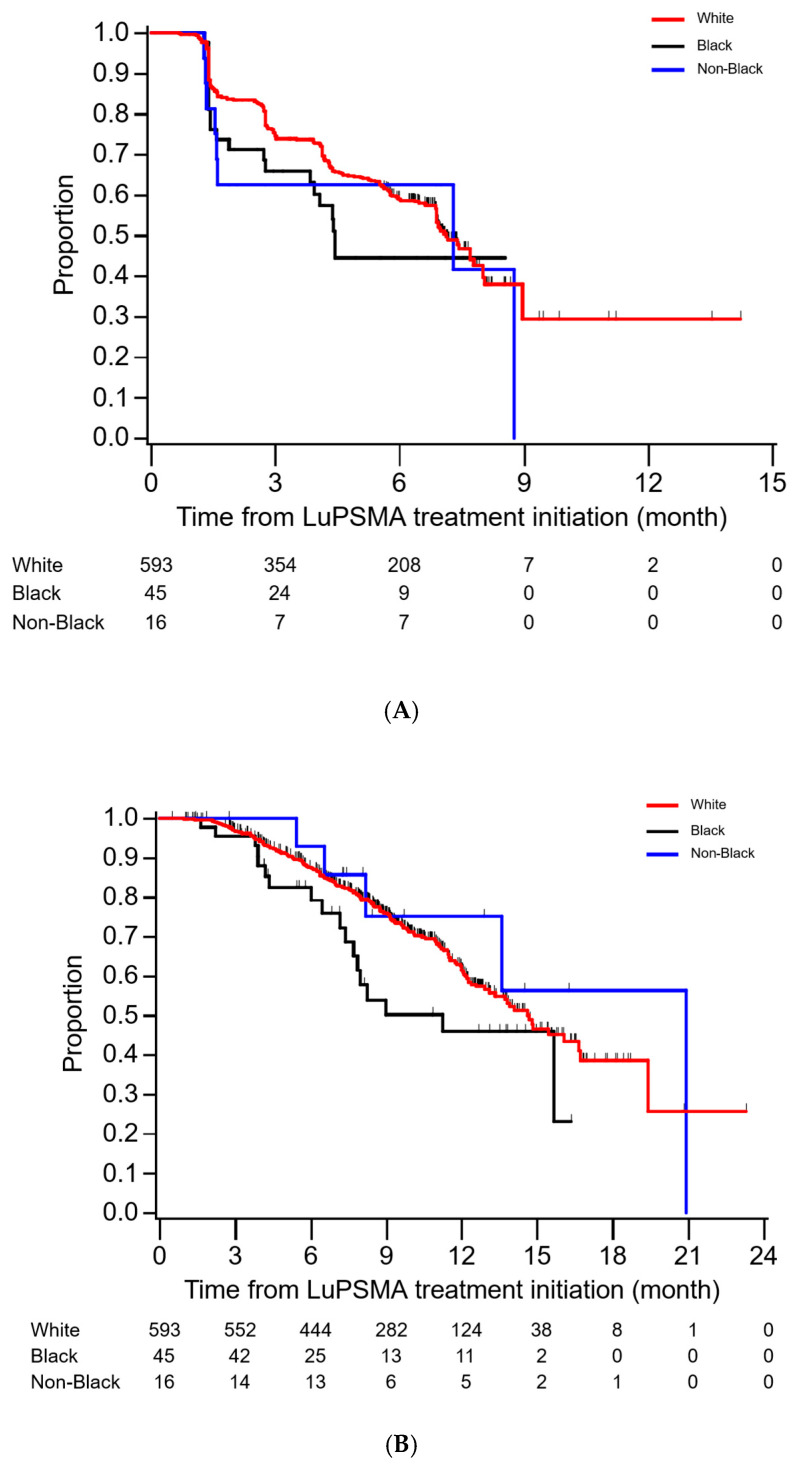
Kaplan–Meier curves showing PSA-PFA (**A**) and OS (**B**), stratified by race.

**Table 1 cancers-17-01960-t001:** Baseline characteristics of included patients, stratified by race.

	Total (N = 654)	White (N = 593)	Black (N = 45)	NBM (N = 16)	*p* (White vs. Black)
Age, median (range)	72 (46–95)	72 (46–95)	70 (51–91)	72 (57–84)	0.32
PSA at C1, ng/mL, median (range)	36 (0–8355)	32 (0–8355)	137 (0–1419)	41 (1–603)	0.01
Number of prior therapies (%), median (range)	4 (2–9)	4 (2–9)	3 (2–7)	4 (2–7)	0.93
2–4	477 (73)	431 (73)	35 (78)	11 (69)	
5	109 (17)	101 (17)	5 (11)	3 (19)	
≥6	68 (10)	61 (10)	5 (11)	2 (13)	
Sites of metastases (%)					
Bone	580 (89)	522 (88)	44 (98)	14 (88)	0.05
Lymph node	446 (68)	403 (68)	32 (71)	11 (69)	0.67
Liver	83 (13)	77 (13)	4 (9)	2 (13)	0.64
Lung	85 (13)	77 (13)	5 (11)	3 (19)	0.72
Visceral metastasis (%)	149 (23)	136 (23)	9 (20)	4 (25)	0.65

Abbreviations: NBM—non-black minority (Asian: n = 8; Hispanic: n = 8); PSA—prostate-specific antigen; C1—cycle 1.

**Table 2 cancers-17-01960-t002:** PSA-progression free survival and overall survival with [^177^Lu]Lu-PSMA-617 in White, Black, and Non-black minority patients.

	White	Black	NBM (%)
**PSA–PFS**			
No. events/N	241/593	21/45	8/16
Median, months (95% CI)	7.1 (6.9–7.8)	4.4 (2.8–NR)	7.3 (1.5–8.8)
6-month PFS rate, % (95% CI)	59 (54–63)	44 (30–63)	62 (39–85)
HR (95% CI)	ref	1.38 (0.89–2.16)	1.28 (0.63–2.60)
*p*-value		0.15	0.49
Adjusted HR *(95% CI)	ref	1.14 (0.72–1.81)	1.16 (0.57–2.35)
Adjusted *p*-value		0.58	0.68
**OS**			
No. events/N	184/593	18/45	5/16
12-month OS rate, % (95% CI)	61 (56–77)	46 (29–66)	75 (47–95)
HR (95% CI)	ref	1.54 (0.95–2.50)	0.80 (0.32–1.97)
*p*-value		0.08	0.62
Adjusted HR * (95% CI)	ref	1.37 (0.84–2.23)	0.68 (0.27–1.74)
Adjusted *p*-value		0.21	0.42
Median follow-up time in alive patients, months (IQR)	9.2 (6.9–12.0)	6.8 (4.1–13.7)	8.0 (7.3–12.9)

Abbreviations: NBM—non-black minority; NR—not reached; HR—hazard ratio; PFS—progression-free survival; CI—confidence interval; OS—overall survival; IQR—interquartile range; ref—referent; * Adjusted hazard ratios were estimated after adjusting for age at treatment initiation, number of prior systemic therapies, sites of metastasis, and PSA levels at C1.

## Data Availability

The data supporting the findings of this study are not publicly available due to patient privacy and institutional restrictions. De-identified data may be made available from the corresponding author upon reasonable request and with appropriate institutional approvals.

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
