# Peer review of "Race and Outcomes to [177Lu]Lu-PSMA-617 in Advanced Prostate Cancer"

_cancers, 2025, doi:10.3390/cancers17121960_

Round 1
Reviewer 1 Report
Comments and Suggestions for Authors
The paper “Race and Outcomes to 177Lu-PSMA-617 in Advanced Prostate Cancer” compares outcomes of LuPSMA therapy among groups of self-reported race: White (N=593), Black (N=45), and non-Black minorities (NBM, N=16)) from 7 institutions across the United States. The outcomes PSA-50 rate, PSA-progression-free survival, and overall survival are evaluated; however, the NBM group is too small for separate evaluation. The study finds only relatively minor and statistically non-significant differences White vs. Black (and similar for White vs. non-White), concluding that LuPSMA has similar efficacy for Black and White patients and thus offers equal treatment benefit across racial groups.
The paper has a solid presentation and analysis, and gives valuable contribution to the area of evaluating medical efficacy in broader radial groups. The Black group is rather small (and even more so the NBM group), but this reflects the available data (pooled from seven institutions), and the authors are aware of the limitations. Conclusions are based upon what the data can support.
This reviewer has only a few minor and typographical comments.
MINOR
In line 106, and similarly in Table 2, the PSA-PSF 95% CI is reported as “[2.8-NR]”. It is unclear to this reviewer how “NR” should be understood. Please clarify or correct.
Regarding notation for radiopharmaceuticals, it is suggested to write “[177Lu]Lu-PSMA-617” (with “177” as superscript) rather than “177Lu-PSMA-617”. As “LuPSMA” is introduced early as abbreviation, this change will only affect a limited number of places in the paper (including the title). The suggestion is based upon a widely accepted consensus paper with guidelines for radiochemistry nomenclature that was published in 2017 and summarized in a widely published open letter that can be found for instance here: https://doi.org/10.1186/s41181-018-0047-y . The European Association of Nuclear Medicine (EANM) has abbreviated it to a short guideline on their web page: https://eanm.org/publications/guidelines/overview/nomenclature/ According to the consensus/guideline, radiopharmaceuticals should be written with the full molecule, including the atom used for labeling (in this case, the full molecule is Lu-PSMA-617), with the specific isotope in square parentheses to the left of this, with no space or hyphen in between.
Supplementary Table 1 appears to be identical to Table 1 in the main paper. If this is indeed the case, it is suggested to omit it from the Supplementary Material. With renumbering, this will leave Table S1 as a reflection of Table 1 (with different stratification) and Table S2 as a reflection of Table 2.
TYPOGRAPHICAL
Line 51: Radium -> radium (names of the elements are not usually capitalized when written out in full)
Lines 77 and 95: Please add a space between number and unit, e.g. 2ng/mL -> 2 ng/mL Similarly for 95%CI -> 95% CI in Table 1 (entry 6-months PFS rate).
Line 82: Chi-square -> chi-square (“chi” is just a Greek letter, not e.g. the name of a person)
C1D1 or C1? In Table 1, the abbreviation C1 is used, but in the table legend it is written as C1D1 – cycle 1. The abbreviation C1D1 is also used in the legend to Table 2 and in the Supplementary Material. Please be consistent (and if C1D1 is kept, please give an indication of why “D1”).
It is suggested to add a page shift (Ctrl+Enter in most word processor programs) before Table 2 to avoid splitting the table over two sides. Given that there is currently space below Figure 1A, it will not even cause extra blank space in the paper as a whole.
Range with comma: In both Table 1 (Number of prior therapies) and Table 2 (Median follow-up time), a range is given with comma, e.g. “(2, 9)”. It is suggested to use a dash instead, “(2-9)”, as is already done for other ranges.
In reference [3] in the reference list, should “(177)Lu” be “177Lu” with the number in superscript?
Author Response
The paper “Race and Outcomes to 177Lu-PSMA-617 in Advanced Prostate Cancer” compares outcomes of LuPSMA therapy among groups of self-reported race: White (N=593), Black (N=45), and non-Black minorities (NBM, N=16)) from 7 institutions across the United States. The outcomes PSA-50 rate, PSA-progression-free survival, and overall survival are evaluated; however, the NBM group is too small for separate evaluation. The study finds only relatively minor and statistically non-significant differences White vs. Black (and similar for White vs. non-White), concluding that LuPSMA has similar efficacy for Black and White patients and thus offers equal treatment benefit across racial groups.
The paper has a solid presentation and analysis, and gives valuable contribution to the area of evaluating medical efficacy in broader radial groups. The Black group is rather small (and even more so the NBM group), but this reflects the available data (pooled from seven institutions), and the authors are aware of the limitations. Conclusions are based upon what the data can support.
This reviewer has only a few minor and typographical comments.
MINOR
In line 106, and similarly in Table 2, the PSA-PSF 95% CI is reported as “[2.8-NR]”. It is unclear to this reviewer how “NR” should be understood. Please clarify or correct.
RESPONSE: We thank the reviewer for this helpful suggestion. We have updated the footnote of Table 2 to include: “NR = not reached.”
Regarding notation for radiopharmaceuticals, it is suggested to write “[177Lu]Lu-PSMA-617” (with “177” as superscript) rather than “177Lu-PSMA-617”. As “LuPSMA” is introduced early as abbreviation, this change will only affect a limited number of places in the paper (including the title). The suggestion is based upon a widely accepted consensus paper with guidelines for radiochemistry nomenclature that was published in 2017 and summarized in a widely published open letter that can be found for instance here: https://doi.org/10.1186/s41181-018-0047-y . The European Association of Nuclear Medicine (EANM) has abbreviated it to a short guideline on their web page: https://eanm.org/publications/guidelines/overview/nomenclature/ According to the consensus/guideline, radiopharmaceuticals should be written with the full molecule, including the atom used for labeling (in this case, the full molecule is Lu-PSMA-617), with the specific isotope in square parentheses to the left of this, with no space or hyphen in between.
RESPONSE: We appreciate the reviewer’s thoughtful suggestion and reference to the consensus guidelines on radiopharmaceutical nomenclature. In accordance with the 2017 EANM-endorsed recommendations, we have updated the manuscript to use the notation “[¹⁷⁷Lu]Lu-PSMA-617” in the title and throughout the text where appropriate. As “LuPSMA” is introduced early as an abbreviation, we have retained its use in the body of the manuscript to ensure readability and consistency.
Supplementary Table 1 appears to be identical to Table 1 in the main paper. If this is indeed the case, it is suggested to omit it from the Supplementary Material. With renumbering, this will leave Table S1 as a reflection of Table 1 (with different stratification) and Table S2 as a reflection of Table 2.
RESPONSE: We thank the reviewer for catching this error. Supplementary Table 1 was identical to Table 1 and has been removed from the supplementary material.
TYPOGRAPHICAL
Line 51: Radium -> radium (names of the elements are not usually capitalized when written out in full)
Lines 77 and 95: Please add a space between number and unit, e.g. 2ng/mL -> 2 ng/mL Similarly for 95%CI -> 95% CI in Table 1 (entry 6-months PFS rate).
Line 82: Chi-square -> chi-square (“chi” is just a Greek letter, not e.g. the name of a person)
C1D1 or C1? In Table 1, the abbreviation C1 is used, but in the table legend it is written as C1D1 – cycle 1. The abbreviation C1D1 is also used in the legend to Table 2 and in the Supplementary Material. Please be consistent (and if C1D1 is kept, please give an indication of why “D1”).
It is suggested to add a page shift (Ctrl+Enter in most word processor programs) before Table 2 to avoid splitting the table over two sides. Given that there is currently space below Figure 1A, it will not even cause extra blank space in the paper as a whole.
Range with comma: In both Table 1 (Number of prior therapies) and Table 2 (Median follow-up time), a range is given with comma, e.g. “(2, 9)”. It is suggested to use a dash instead, “(2-9)”, as is already done for other ranges.
In reference [3] in the reference list, should “(177)Lu” be “177Lu” with the number in superscript?
RESPONSE: Thank you for these detailed suggestions. We have implemented all recommended typographical and formatting changes, including standardizing terminology, correcting spacing and capitalization, and updating radiopharmaceutical notation. We appreciate your close review and believe these edits improve the clarity and consistency of the manuscript.
Reviewer 2 Report
Comments and Suggestions for Authors
I do not think there should have been a reason to think that race affects outcome on Lutetium-PSMA, nor was any hypothesis on why such a discrepance should occur was provided. Indeed, there is no significant difference in outcome, as expected.
I have no methodological concerns, but unfortunately the manuscript is not of sufficient interest or significance.
Author Response
Reviewer 2
I do not think there should have been a reason to think that race affects outcome on Lutetium-PSMA, nor was any hypothesis on why such a discrepance should occur was provided. Indeed, there is no significant difference in outcome, as expected.
I have no methodological concerns, but unfortunately the manuscript is not of sufficient interest or significance.
RESPONSE: We thank the reviewer for their comments and agree that our findings suggest no significant difference in outcomes to LuPSMA by race. However, we respectfully note that disparities in prostate cancer outcomes are well documented in the literature, and Black patients have historically been underrepresented in clinical trials of advanced prostate cancer therapies, including LuPSMA. As noted in our Introduction and Discussion, prior studies have raised the possibility of differential responses by race to some treatments, and we sought to address a critical evidence gap using real-world data. We hope the reviewer will agree that confirming the absence of disparity in a multi-institutional cohort is a valuable contribution to the literature and may support efforts to ensure equitable access to this therapy. We appreciate the reviewer’s time and careful review.
Reviewer 3 Report
Comments and Suggestions for Authors
The authors review and compare the outcomes with radioligand therapy for advanced or refractory prostate cancer by race/ethnicity. Notably in a large retrospective cohort of 654 patients, 593 were white underscoring the message about unequal access to what is now within standard of care (chart review includes up to 2024 patients; NEJM paper for VISION study was 2021). Notably they find that--unsurprisingly--radioligand therapy is as effective for patients who are black or hispanic as it is for patients who are white. This is a valuable contribution to the literature and I recommend acceptance.
A couple minor things that could be incorporated into discussion include mention of how prior work shows similar treatment benefit for localized prostate cancer based on race (PMID 34184271), more equitable access systems such as the VHA in the US show fewer disparities (PMID 33892497), and how delays and lack of access to care remain an issue in the US even with health care reform (PMID 37537835)
Author Response
Reviewer 3
The authors review and compare the outcomes with radioligand therapy for advanced or refractory prostate cancer by race/ethnicity. Notably in a large retrospective cohort of 654 patients, 593 were white underscoring the message about unequal access to what is now within standard of care (chart review includes up to 2024 patients; NEJM paper for VISION study was 2021). Notably they find that--unsurprisingly--radioligand therapy is as effective for patients who are black or hispanic as it is for patients who are white. This is a valuable contribution to the literature and I recommend acceptance.
A couple minor things that could be incorporated into discussion include mention of how prior work shows similar treatment benefit for localized prostate cancer based on race (PMID 34184271), more equitable access systems such as the VHA in the US show fewer disparities (PMID 33892497), and how delays and lack of access to care remain an issue in the US even with health care reform (PMID 37537835)
RESPONSE: We thank the reviewer for these thoughtful suggestions. In response, we have revised the Discussion to incorporate findings from the cited studies (page 6, lines 236-243; page 7, lines 244-246).
Reviewer 4 Report
Comments and Suggestions for Authors
Rami et al present a multi-institutional retrospective cohort series of outcomes of patients with mCRPC treated with LuPSMA with a focus on outcomes based upon self identified race. The background for this is differences in survival and response based upon self identified AA vs other for PCa. In general, they show similar response rates. Although the total numbers are small, this adds to literature given the paucity of data previously. No other outstanding issues identified in the manuscript.
Author Response
Reviewer 4
Rami et al present a multi-institutional retrospective cohort series of outcomes of patients with mCRPC treated with LuPSMA with a focus on outcomes based upon self identified race. The background for this is differences in survival and response based upon self identified AA vs other for PCa. In general, they show similar response rates. Although the total numbers are small, this adds to literature given the paucity of data previously. No other outstanding issues identified in the manuscript.
RESPONSE: We thank the reviewer for their positive assessment of our manuscript and for recognizing the contribution of this work to the existing literature.